# Spatio-Temporal Differentiations and Influence Factors in China's Grain Supply Chain Resilience

**Jinrui Chang**  **and Huiming Jiang** *

College of Economics and Management, Jilin Agricultural University, Changchun 130118, China; changjinrui@mails.jlau.edu.cn
* Correspondence: jhm573@163.com; Tel.: +86-0431-84533071

**Abstract:** Based on Chinese provincial panel data (2011–2020), the CRITIC-EWM, kernel density estimation, Dagum Gini coefficient and a spatial Dubin model were used for analyzing endogenous and exogenous factors to enhance grain supply chain resilience. The results illustrated the following: (1) Ensuring the resilience of the grain supply chain's ability to digest, recover and learn was fundamental to internal system robustness. (2) Overall, China's grain supply chain resilience increased year over year, but the growth rate was low and showed an attenuation trend of east-middle-west. (3) There was a significant spatial positive correlation among provinces and grain supply chain resilience, however, the large gap and slow development led to the formation of polarization with high-high and low-low clusters. (4) The exogenous drivers were all significant and had a significant positive spillover effect on the grain supply chain resilience. (5) Subregional regressions reflected the heterogeneity of the influence factors, which highlighted the implementation of targeted strategies as the key to achieving synergy.

**Keywords:** grain supply chain; resilience; spatio-temporal differentiation; spatial Dubin model; heterogeneity



## 1. Introduction

### 1.1. Background

In recent years, the spread of the COVID-19 pandemic, natural disasters and geopolitical conflicts have hampered the path to food security with Chinese characteristics. Differences in resource endowments and overdraft of the agricultural ecological environment have also restricted high-quality development of food [1]. As a consumption necessity, grain shows stronger sensitivity and vulnerability in dealing with emergencies such as uncertainty and risk [2]. It is very important to resist a crisis and to recover quickly from a crisis [3]; continuously improving the resilience of the grain supply chain is the key to consolidating the foundation of food security.

During the COVID-19 pandemic, the vulnerability of national agri-food systems became evident, and this was highlighted by the UN Food and Agriculture Organization (FAO). In September 2021, the UN Food System Summit focused, among other goals, on building resilience to vulnerabilities, shocks and stresses to ensure the continued operation of healthy and sustainable food systems. Currently, in China, grain loss in post-production links such as storage, transportation and processing has reached more than 70 billion jin every year [4]. The increasing loss and waste of grains reduce food security and also negatively affect sustainable development [5]. Meanwhile, the links of grain production, purchase, storage, processing, transportation and consumption are loosely connected and the circulation efficiency is low [6]. The trend of "non-agriculturalization" and "non-foodization" of grain production is obvious, and the rural labor force is shifting to the non-agricultural sector. Under a complex international and domestic environment, stabilizing domestic grain self-sufficiency, ensuring the absolute security of grain and continuously

promoting development of the grain supply chain based on the integration of "production, purchase, storage, processing and consumption" is a prerequisite [7]. However, there are questions related to how to ensure a sustainable supply of domestic grain in China, how to effectively compensate for China's grain supply chain losses, and how to optimize China's grain supply chain resilience. In this paper, we aim to answer these questions by constructing a grain supply chain resilience evaluation system to objectively analyze China's provincial grain supply chain resilience and study the regional spatio-temporal differences, and therefore, help to promote high-quality synergistic development and to explain a feasible way to efficiently strengthen China's grain supply chain resilience based on macroscopic exogenous drivers.

### 1.2. Literature Review

Sustainability of a system under uncertainty is similar to the definition of resilience. Resilience was first proposed and defined by Holling as the tendency of a system to maintain its organizational structure and productivity after being disturbed [8]. Supply chain resilience is based on the research foundation of ecosystems, economics and risk management [9]. The concept of supply chain resilience combines these previous tenets with the study of supply chain vulnerability, which has been defined by Svensson as unexpected deviations from the norm and their negative consequences [10]. The definition of grain supply chain resilience evolved from the definition of resilience, agricultural resilience and food system resilience. According to Folke [11], agricultural resilience can be interpreted as the ability of an agricultural system to ensure that its original characteristics are not erased and key functions are not lost in the face of objective external disturbances such as natural disasters, policy orientations and market changes. Hao Aimin and others described food system resilience as the ability of a food system to resist external shocks, to recover quickly from shocks and to shift to new growth paths for adaptive development through internal organizational restructuring of the system [12]. As an overview of the concepts, we define resilience of a grain supply chain as the ability to maintain and to restore continuous operation of the main functions of a chain system connected by various economic subjects related to the development of the grain industry after being disturbed and impacted, and to stabilize the functions of production, purchasing, storage, processing and consumption within the grain industry.

In addition, scholars have conducted thorough research on how to evaluate resilience. Fiksel proposed four major characteristics of resilient systems: diversity, efficiency, adaptability and cohesion [13]. Martin believed that the four dimensions of resilience for dealing with recession and impact included prevention, recovery, renewal and redirection [14]. The IPCC has emphasizes that resilience should also maintain the ability to adapt, learn and transform [15]. The FAO pointed out that a truly resilient agricultural food system must have solid prevention, prediction, digestion, adaptation and transformation abilities, and must resist any destructive factors [16]. Fan Xuemei and Lu Mengyuan evaluated the supply chain resilience of Chinese automotive companies during the COVID-19 pandemic based on prediction, reaction, adaptation, recovery and learning ability [17]. Liao Han measured the resilience of China's supply chain under adverse impact based on damage degree, recovery degree and recovery time [18]. Zhang Mingdou and others analyzed the spatial differences and influencing factors of China's agricultural economic resilience from two dimensions of resistance and reconstruction [19]. The above studies provide a solid theoretical foundation for this paper, and we think that the more specific the evaluation of system resilience is, the more helpful it is to provide targeted countermeasures. In this study, grain supply chain resilience is measured based on prevention, prediction, digestion, recovery, learning and transformation.

It is worth emphasizing that the studies on grain supply chain resilience are relatively inadequate, with foreign scholars focusing more on the optimization of micro instances and domestic scholars mainly focusing on qualitative research on mechanisms and theories. Thus, in this paper, we focus on domestic grain supply to calculate grain supply chain

resilience. The theoretical value is the deconstruction of the grain supply chain system by main functions and the visualization of resilience capacity dimensions, which improves the accuracy of the system resilience evaluation, clarifies the weaknesses of the internal system resilience and enhances the targeting of the research. The empirical value is that the study is based on Chinese provinces, and the analysis of spatial and temporal characteristics can provide suggestions for synergistic sustainable development, can introduce external drivers to measure effective impact scenarios, and can provide references for strengthening the implementation of regional grain supply chain resilience policies.

## 2. Materials and Methods

### 2.1. Research Design

By deconstructing the grain supply chain into subfunctional links of grain production, storage, processing, transportation and consumption, the system resilience evaluation system is formed by combining prevention, prediction, digestion, recovery, learning and transformation. The strategy of strengthening system resilience is analyzed from the perspective of exogenous driving factors such as factor quality, structural upgrading, institutional environment and economic environment. The research framework is shown in Figure 1.

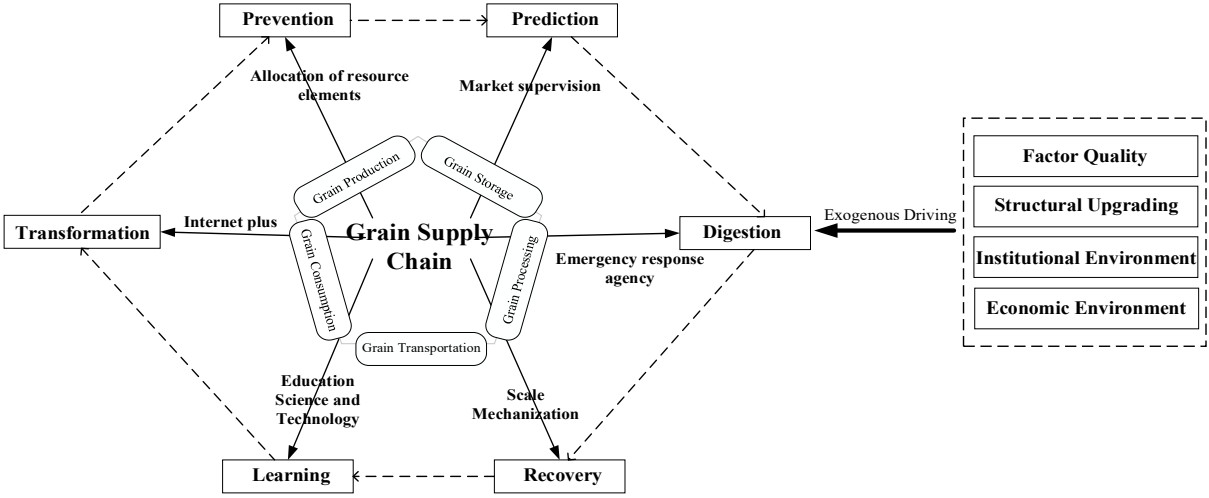

**Figure 1.** Research logical framework.

Figure 1 shows that all links of the system cooperate with each other, and each capability dimension progressively forms a closed loop to evaluate the resilience of the system. Considering that resilience itself is the robustness of system structure under the impact of uncertainty, the exogenous driving factors of grain supply chain resilience are factor quality, structural upgrading, institutional environment and economic environment. As a result, this study is more specific and forms a two-dimensional internal and external scenario for improving grain supply chain resilience.

### 2.2. Construction of a Grain Supply Chain Resilience Evaluation Index System

According to the analysis, the evaluation index system for measuring the resilience of a grain supply chain is constructed as shown in Table 1.

**Table 1.** The index system for evaluating the resilience of grain supply chain.

| First-Class Index | Second-Class Index | Index Interpretation | Target Vector |
|---|---|---|---|
| Prevention | Grain [1] sown area index | Year-on-year change rate of grain sown area | + |
| | Effective irrigation rate of grain | Grain effective irrigated area/grain sown area | + |
| | Volatility of grain purchase | Year-on-year growth rate of grain purchase by state-owned enterprises | + |
| | Number of national key leading enterprises in agricultural industrialization | Supervise the number of qualified national key leading enterprises in agricultural industrialization | + |
| | Productivity of main products of grain processing enterprises | Quantity of main products produced by grain processing enterprises per unit | + |
| | Road density in the jurisdiction | Ratio of road density to area in jurisdiction | + |
| | Direct consumption of grain per capita | Total grain consumption and population ratio in urban and rural areas | − |
| Prediction | Grain yield per unit area | Grain yield per hectare per unit area | + |
| | Ratio of affected area | Proportion of affected area to sown area of grain | − |
| | Number of price monitoring outlets | The sum of grain price monitoring outlets at all levels | + |
| | Grain marketization index | Grain commodity rate | + |
| Digestion | Grain reserve level | Grain stocks of state-owned grain enterprises/grain output in the province | + |
| | Number of emergency processing enterprises | Number of emergency grain processing enterprises at all levels | + |
| | Number of enterprises in emergency storage & distribution centers | Number of enterprises in emergency warehousing and distribution centers | + |
| | Number of emergency supply network points | Emergency supply outlets at all levels guarantee market food supply outlets | + |
| Recovery | Multiple cropping index | Total area of grain sown (or transplanted)/total area of cultivated land in the whole year | + |
| | Mechanization level of grain planting | Total power of agricultural planting machinery at the end of the year × a [1] | + |
| | Grain labor productivity | Ratio of total grain output to rural labor force × b [1] | + |
| | Annual production and processing capacity of grain processing industry | Year-on-year volatility of total annual processed grain output of various grain processing enterprises | + |
| Learning | Entrustment of new varieties of agricultural plants | Number of R&D authorizations for new agricultural varieties per year | + |
| | Education level of rural residents | Average years of education of rural residents | + |
| | Ratio of scientific research personnel among grain employees | Regional grain output value/regional GDP × regional R&D full time | + |
| Transformation | Enterprise e-commerce coverage rate | Proportion of enterprises with e-commerce transactions in total enterprises | + |
| | E-commerce development index | The proportion of e-commerce transaction volume to GDP | + |

[1] According to the China Statistical Yearbook, grain refers to grain except grains, potatoes and beans. Drawing lessons from Wang Ruifeng [20] and other related research, a = grain sown area/crop sown area, b = grain (raw grain) output value/agricultural output value, which are targeted to characterize grain-related index data.

This research was conducted by summarizing the panel data of 31 provinces, municipalities and autonomous regions (except Hong Kong, Macao and Taiwan) from 2011 to 2020. The data sources were as follows: (1) grain sowing area index, effective irrigation rate of grain, road density of jurisdictions, disaster area, number of rural labor force were obtained from the *China Rural Statistical Yearbook*; (2) the number of supervised and qualified national key leading enterprises of agricultural industrialization was obtained from the website of the Ministry of Agriculture and Rural Affairs of the People's Republic of China; (3) grain output value, grain purchases and stocks of state-owned enterprises, grain yields, grain reserve levels, grain cultivation mechanization levels, grain labor productivity, and annual production and processing capacity of the grain processing industry were obtained from the *China Grain and Materials Reserve Yearbook*; (4) Price monitoring network points, emergency processing outlets, the number of emergency storage, the number of emergency distribution center enterprises, and emergency supply outlets were obtained from the provincial food work in the *China Grain and Material Reserve Yearbook*; (5) the number of authorized new agricultural plant variety rights was obtained from the *China Science and Technology Statistical Yearbook*; (6) the education levels of rural residents were obtained from the *China Population and Employment Statistical Yearbook*; (7) the arable land area, regional R&D all-time volume, enterprises of e-commerce trading activities and e-commerce transaction amount were obtained from the *China Statistical Yearbook*; (8) the grain commodity rate and the main product productivity of grain processing enterprises were obtained from the Brice Agricultural Database, meanwhile, the main product productivity of grain processing enterprises was mainly accounted by the productivity of rice, wheat and grain oil. Among the data, individual index data could not be obtained directly, and therefore, we applied the interpolation method and linear programming method to supplement some vacant data.

*2.3. Methods*

2.3.1. CRITIC-EWM Combined Evaluation

CRITIC-EWM combines the CRITIC weight evaluation method and the entropy weight method. Considering the correlation degree among the indexes of the components of the grain supply chain system, in order to fully select the objective attributes of the data for scientific evaluation and to avoid the disadvantages of potential data information loss in the original single weight calculation method, in this paper, we applied the CRITIC-EWM combined weight evaluation method [21] to improve the accuracy of index weighting and evaluation.

Assuming there are $n$ samples and $p$ indexes, $x_{ij}$ is the $j$ index value of the $i$ sample.

- Regarding standardization treatment of indicators, because different indicators have different dimensions and properties, it is necessary to carry out standardization treatment. According to the different properties of each index, positive and negative standardization treatment was carried out: Positive standardization: $x'_{ij} = \frac{x_{ij} - x_{\min}}{x_{\max} - x_{\min}}$ Negative standardization: $x'_{ij} = \frac{x_{\max} - x_{ij}}{x_{\max} - x_{\min}}$
- The CRITIC weight method is used to determine weight.

Calculate the information amount $C_j$ of a single indicator as follows:

$$C_j = S_j \sum_{i=1}^{n} (1 - r_{ij}) = S_j \times R_j$$
$$\omega_{1j} = \frac{C_j}{\sum_{j=1}^{p} C_j} \tag{1}$$

where $\omega_{1j}$ is the weight of the first index, $C_j$ is the information amount of a single index, and $S_j$ is the standard deviation of the $j$ index. The standard deviation is used to express the difference and fluctuation of the internal values of each index. The larger the standard deviation, the greater the numerical difference of the index, the more information it can show and the stronger the evaluation intensity. $R_j$ is the correlation coefficient and $r_{ij}$

represents the correlation coefficient between evaluation indexes *i* and *j*, which is used to reflect the correlation between indexes. The stronger the correlation with other indexes, the smaller the conflict between the index and other indexes, and the more the same information is reflected, the smaller the distribution weight of the index should be.

- The EWM determines weight, and because of its wide application, we will not repeat it in this paper.
- Calculate the CRITIC-EWM combined weight $\lambda_j$ and the comprehensive score $R_i$ of the resilience of grain supply chain in each province as follows:

$$\begin{aligned} \lambda_j &= \rho\omega_{1j} + (1-\rho)\omega_{2j} \\ R_i &= \sum_{j=1}^{p} \lambda_j x'_{ij} \end{aligned} \tag{2}$$

Because of the equivalence between the CRITIC weight method and the entropy weight method, $\rho = 0.5$.

### 2.3.2. Kernel Density Estimation

Kernel density estimation [22] is a nonparametric estimation method. In this paper, Gaussian kernel function kernel density estimation is used to dynamically analyze the spatial distribution of grain supply chain resilience in each region, during the study period. The density function of the random variable *X* is:

$$f(x) = \frac{1}{NH}\sum_{i=1}^{N} K(\frac{X_i - x}{H}) \tag{3}$$

where $X_i$ represents the observed value of independent and identical distribution, *X* represents the mean value, *H* denotes broadband, *N* represents the number of observed values, and *K* denotes nuclear density.

### 2.3.3. Dagum Gini Coefficient

Dagum proposed the Gini coefficient method according to subgroup decomposition, which can decompose the source of regional differences as well as evaluate interactions between individuals, making up for the limitations of traditional measurement methods of regional differences. In this paper, we draw lessons from the application methods of Xu Xiaoxin [23], and its expression is as follows:

$$G = \frac{\sum_{j=1}^{k}\sum_{h=1}^{k}\sum_{i=1}^{n_j}\sum_{r=1}^{n_h}\left|y_{ji} - y_{hr}\right|}{2n^2\overline{y}} \tag{4}$$

Among them, the Dagum Gini coefficient decomposes the total Gini coefficient into intra-regional and inter-regional difference contributions and super-variable density contribution. Here, *n* represents the total number of provinces, *k* represents the total number of regions, *j* and *h* represent different regions, *i* and *r* represent different provinces, $n_j$ ($n_h$) is the number of provinces within the *j* (*h*) region, $y_{ji}$ ($y_{hr}$) is the resilience index of grain supply chain of provinces *i* (*r*) in *j* (*h*) region, and $\overline{y}$ represents the average value of grain supply chain resilience of each province.

### 2.3.4. Spatial Autocorrelation

- Global spatial autocorrelation

The global Moran's I [24] was measured to judge the spatial correlation degree on the geographical distance of the study samples, and its value range is $[-1, 1]$. A positive value indicates that there is positive spatial correlation, a negative value indicates that there

is negative spatial correlation, and the greater the absolute value, the greater the spatial correlation, and vice versa.

- Local spatial autocorrelation

To investigate the agglomeration distribution of local areas in space, according to Moran's I, the study samples were divided into four types: high-high agglomeration area, low-low agglomeration area, high-low agglomeration area and low-high agglomeration area, so as.

### 2.3.5. Spatial Dubin Model

A spatial Dubin model [25] can investigate the endogenous correlation of dependent variables, can identify the direct and indirect effects of external factors, and can accurately estimate the spatial correlation and the degree of action of influencing factors. An SDM decomposition model can better reflect the spillover effect of geographical elements, including direct, indirect and total effect models. A direct effect model measures the effect of independent variables on the dependent variables in this region, while an indirect effect model measures the effect of independent variables on the dependent variables in related regions; the sum of the two models is the total effect. This method was introduced to analyze the influence and spatial spillover effect of the exogenous driving factors of grain supply chain resilience.

## 3. China's Grain Supply Chain Resilience

### 3.1. Combined Weights of Each Index for Grain Supply Chain Resilience

The CRITIC-EWM results for the combined weight of each index for the grain supply chain are shown in Table 2.

**Table 2.** The combined weight of each evaluation index.

| First-Class Index | Second-Class Index | CRITIC Weight | EWM Weight | Combined Weight |
|---|---|---|---|---|
| Prevention | Grain sown area index | 0.0291 | 0.0026 | 0.0158 |
| | Effective irrigation rate of grain | 0.0533 | 0.0298 | 0.0415 |
| | Volatility of grain purchase | 0.0341 | 0.0110 | 0.0225 |
| | Number of national key leading enterprises in agricultural industrialization | 0.0372 | 0.0242 | 0.0307 |
| | Productivity of main products of grain processing enterprises | 0.0470 | 0.0585 | 0.0528 |
| | Road density in the jurisdiction | 0.0506 | 0.0297 | 0.0401 |
| | Direct consumption of grain per capita | 0.0422 | 0.0114 | 0.0268 |
| Prediction | Grain yield per unit area | 0.0474 | 0.0628 | 0.0551 |
| | Ratio of affected area | 0.0296 | 0.0703 | 0.0500 |
| | Number of price monitoring outlets | 0.0213 | 0.1582 | 0.0897 |
| | Grain marketization index | 0.0411 | 0.0515 | 0.0463 |
| Digestion | Grain reserve level | 0.0421 | 0.0610 | 0.0516 |
| | Number of emergency processing enterprises | 0.0429 | 0.0373 | 0.0401 |
| | Number of enterprises in emergency storage & distribution centers | 0.0475 | 0.0151 | 0.0313 |
| | Number of emergency supply network points | 0.0532 | 0.0590 | 0.0561 |
| Recovery | Multiple cropping index | 0.0404 | 0.0462 | 0.0433 |
| | Mechanization level of grain planting | 0.0489 | 0.0623 | 0.0556 |
| | Grain labor productivity | 0.0341 | 0.0812 | 0.0577 |
| | Annual production and processing capacity of grain processing industry | 0.0297 | 0.0042 | 0.017 |
| Learning | Entrustment of new varieties of agricultural plants | 0.0616 | 0.0295 | 0.0455 |
| | Education level of rural residents | 0.0490 | 0.0323 | 0.0407 |
| | Ratio of scientific research personnel among grain employees | 0.0393 | 0.0527 | 0.046 |
| Transformation | Enterprise e-commerce coverage rate | 0.0327 | 0.0036 | 0.0182 |
| | E-commerce development index | 0.0457 | 0.0053 | 0.0255 |

According to the calculation results, the level of the grain reserves is crucial, with a weight of 0.0897. In addition, it is highlighted that guaranteeing grain supply chain resilience must rely on the authorization of new agricultural plant variety rights, the level of mechanization of grain cultivation, the annual production and processing capacity of the grain processing industry, the number of price monitoring network points, the productivity of the main products of grain processing enterprises, and the number of enterprises in emergency storage and distribution centers, which fully reflects an effective strategy to strengthen resilience. The foundation of grain supply chain resilience must ensure the "quantity" of food security, inject innovation-driven elements and drive the supply side to cater to market changes in a timely and sustainable manner.

### 3.2. Temporal and Spatial Evolution Characteristics

The provinces in China were divided into four echelons by using the natural breakpoint method by Arcgis10.8; the four echelons were used to target areas with strong resilience, sufficient resilience, medium resilience and weak resilience, as shown in Figure 2.

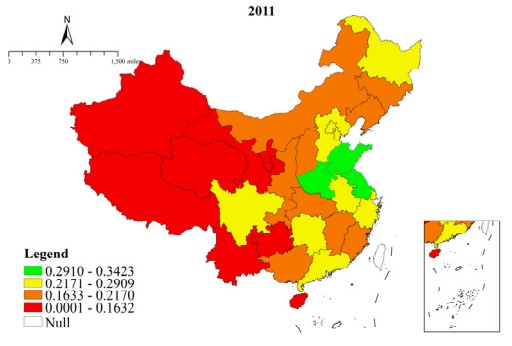

(**a**) China's grain supply chain resilience in 2011

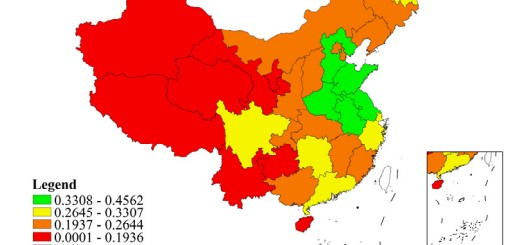
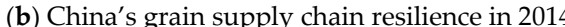

(**b**) China's grain supply chain resilience in 2014

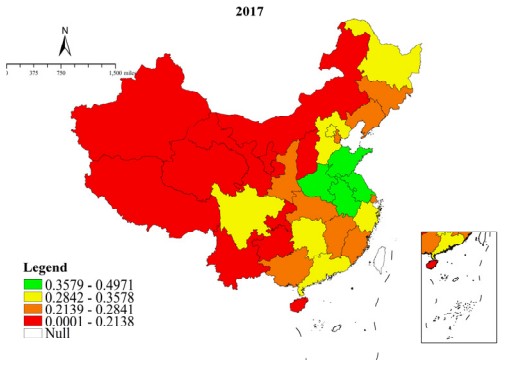

(**c**) China's grain supply chain resilience in 2017

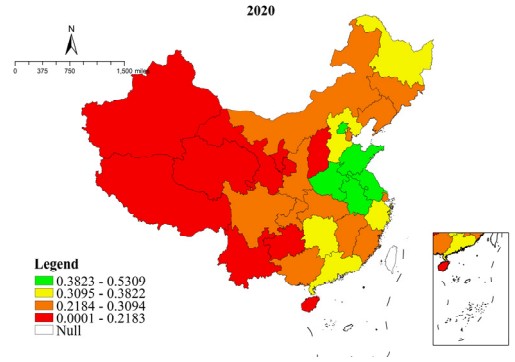

(**d**) China's grain supply chain resilience in 2020

**Figure 2.** The spatio-temporal evolution of China's grain supply chain resilience in 2011–2020.

According to the temporal and spatial evolution, in the time dimension, the resilience index of China's provincial grain supply chain is increasing year over year, but the values are still low, indicating that the grain supply chain still needs to be strengthened. In 2020, the average value of China's resilience index was 0.2893, which was less than 1% higher than that in 2019. In the spatial dimension, there is a trend of recession in the east-middle-west. During the study period, except Shandong, the rankings of provinces fluctuate frequently, which indicates that the stability of China's grain supply chain is weak at present. The strong resilience echelons includes Shandong, Jiangsu, Anhui and Henan; the sufficient resilience echelons include Hebei, Heilongjiang, Beijing, Guangdong, Hunan, Zhejiang and Sichuan; the medium resilience echelons include Shanghai, Tianjin, Hubei, Jiangxi, Liaoning, Fujian, Shaanxi, Jilin, Guangxi, Inner Mongolia, Chongqing, Shanxi

and Guizhou; the weak resilience echelons include Gansu, Xinjiang, Yunnan, Hainan, Ningxia, Tibet and Qinghai. By comparing the growth rates, it is found that the resilience of Beijing, Zhejiang, Guangdong, Shanghai and Tianjin, which are the main grain sellers, is growing rapidly. It proves that the main grain producing areas have strong neighboring communication on the grain supply side, driving and radiating high-quality development of the grain industry in neighboring provinces, and fully demonstrating that ensuring grain production capacity is the key to enhancing resilience. In addition, in the areas where production advantages are compared, it is found that the grain supply chain resilience in the three northeastern provinces does not match the production capacity, reflecting the lack of digestion, recovery, learning and transformation abilities in the development of the grain industry in recent years. Therefore, functional provinces, as major grain producers, must vigorously consolidate grain supply chain resilience, otherwise, they will easily expose their vulnerability under the impact of uncertainty and weaken the stability of food security in space.

### 3.3. Regional Spatial Distribution Characteristics

According to the measurement of grain supply chain resilience, it is more effective to take the northeast, eastern, middle and western China as spatial analysis units [26], which is conducive to fully interpreting spatial differences. The distributions of the nuclear density curves in each region from 2011 to 2020 are shown in Figure 3.

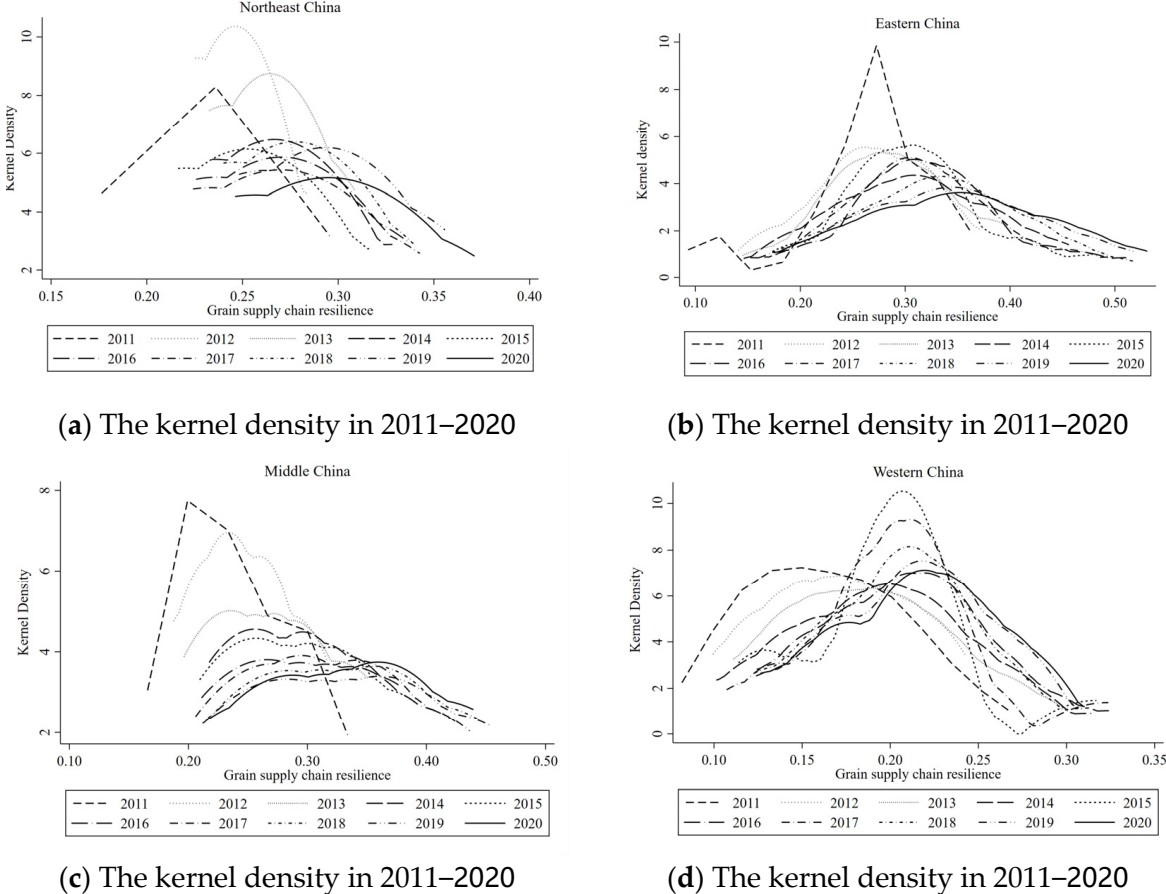

(**a**) The kernel density in 2011–2020

(**b**) The kernel density in 2011–2020

(**c**) The kernel density in 2011–2020

(**d**) The kernel density in 2011–2020

**Figure 3.** The kernel density curves in four spatial units.

In the Figure 3a–c, we could conclude that the distributions of northeast China, eastern China and middle China are similar as a whole, and the spatial characteristics are flattening and gradual to the right. The "peak" height decreases year over year, gradually tends to a single peak, moves to the right and broadens the distribution ductility, which explains

that grain supply chain resilience in northeast China, eastern China and middle China is constantly increasing. However, the number of advantageous provinces is reduced, and the evolution distribution is concentrated, which exposes the problem of development block. However, only the eastern region has an obvious tailing phenomenon and no dynamic convergence, which shows that the internal spatial gap is expanding and deepening. Combined with Figure 2, it is concluded that the slow development of weak and resilient provinces is serious.

In the Figure 3d, the nuclear density curve in western China is quite different from the other three regions. In 2015, the "counterattack point" was ushered in. Meanwhile, the maximum and minimum values of the nuclear density curve coexisted during the research period, indicating that there were huge differences between the poles in the region and the number of provinces with strong and weak resilience was also the largest. There is no elongated right tail in the nuclear density curve, which indicates that the gap between provinces is solidified. Especially, there will be "double peaks" in 2020, which verifies the gradient difference of toughness development within the region.

## 4. Spatial Differences of China's Grain Supply Chain Resilience
### 4.1. Spatial Differences of Regional Grain Supply Chain Resilience

The Dagum Gini coefficient is introduced to calculate the contribution rates of inter-regional and regional differences, as shown in Table 3.

**Table 3.** The space Gini and contribution of the four spatial units' grain supply chain resilience.

| Year | Total | Intra-Regional Differences | | | | Regional Differences | | | | | | Contribution Rate | | |
|------|-------|-----------|---------|--------|---------|-----|-----|-----|-----|-----|-----|----------------|----------|------------------------|
| | | Northeast | Eastern | Middle | Western | N-E | N-C | N-W | E-C | E-W | C-W | Intra-Regional | Regional | Supervariable Density |
| 2011 | 0.1710 | 0.07 | 0.12 | 0.09 | 0.15 | 0.13 | 0.09 | 0.18 | 0.13 | 0.26 | 0.20 | 21.26% | 62.74% | 16.00% |
| 2012 | 0.1668 | 0.05 | 0.13 | 0.10 | 0.15 | 0.12 | 0.08 | 0.18 | 0.13 | 0.24 | 0.19 | 22.83% | 60.16% | 17.01% |
| 2013 | 0.1717 | 0.06 | 0.13 | 0.12 | 0.16 | 0.12 | 0.10 | 0.19 | 0.13 | 0.24 | 0.21 | 23.32% | 55.70% | 20.98% |
| 2014 | 0.1839 | 0.08 | 0.15 | 0.12 | 0.16 | 0.14 | 0.12 | 0.18 | 0.14 | 0.25 | 0.22 | 23.23% | 57.20% | 19.57% |
| 2015 | 0.1760 | 0.09 | 0.13 | 0.12 | 0.14 | 0.15 | 0.14 | 0.15 | 0.14 | 0.24 | 0.22 | 21.75% | 60.33% | 17.92% |
| 2016 | 0.1835 | 0.09 | 0.15 | 0.14 | 0.15 | 0.16 | 0.14 | 0.16 | 0.15 | 0.25 | 0.23 | 22.89% | 56.18% | 20.93% |
| 2017 | 0.1838 | 0.10 | 0.15 | 0.13 | 0.14 | 0.15 | 0.15 | 0.17 | 0.14 | 0.25 | 0.24 | 21.97% | 57.45% | 20.58% |
| 2018 | 0.1874 | 0.08 | 0.15 | 0.13 | 0.14 | 0.15 | 0.14 | 0.17 | 0.15 | 0.26 | 0.25 | 21.30% | 60.55% | 18.15% |
| 2019 | 0.1873 | 0.08 | 0.15 | 0.14 | 0.13 | 0.15 | 0.14 | 0.17 | 0.15 | 0.26 | 0.24 | 21.48% | 60.20% | 18.32% |
| 2020 | 0.1899 | 0.09 | 0.15 | 0.13 | 0.14 | 0.16 | 0.14 | 0.18 | 0.15 | 0.27 | 0.24 | 21.46% | 61.49% | 17.05% |
| AVG | 0.1801 | 0.08 | 0.14 | 0.12 | 0.14 | 0.14 | 0.12 | 0.17 | 0.14 | 0.25 | 0.22 | 22.15% | 59.20% | 18.65% |

The average Gini coefficient of China is 0.1801, which fluctuates up and down at the 1% level, indicating that the spatial difference of China's grain supply chain resilience is solidified. In 2012, the state implemented various grain subsidies, minimum purchase prices and tax exemption policies in circulation with "stable and increased production" as the core, which achieved remarkable results and weakened the spatial differences. After 2015, the overall Gini coefficient increased year over year, and reached a new highest value in 2020, indicating that the bipolar difference of provincial resilience in China is deepening. The difference contribution rate shows that inter-regional differences are the main cause, accounting for 59.2%, while intra-regional differences are the auxiliary cause, accounting for 22.15%. The contribution of super-variable density fluctuates frequently, with an average value of 18.65%, which explains that the grain supply chain resilience in the four spatial units overlaps the strength area and the weakness area. This makes it more difficult to make overall planning to achieve a high degree of adaptation of development within and between regions.

From the results of regional differences, it can be found that the average Gini coefficients among the eastern, middle and western regions are 0.25 and 0.22, and the large cross-regional gap exposes the "vulnerable point" of food security, which verifies the spatial decline of resilience in the eastern, middle and western regions. The differences between northeast China, eastern China and middle China are 0.14 and 0.12, and the lack of resilience in areas with superior production capacity has the potential risk of weakening China's food security. Through the time evolution of regional differences, it shows that

the solidification problem of spatial gap is serious, and the fundamental reason lies in the slow development of weak and resilient areas. On this basis, the demand for high-quality and diversified grain aggravates the effectiveness of the supply-side reform of the grain industry. Over-reliance on various protection policies and the inability of the income from areas with advantageous production capacity to stay in grain producing areas lead to weak production enthusiasm, and the contradiction between external and internal factors leads to a weak foundation for the grain supply chain.

### 4.2. Analysis of Spatial Agglomeration Characteristics

4.2.1. Global Spatial Autocorrelation

Using stata17.0, the global Moran's I (Table 4) of the China's provincial grain supply chain resilience was obtained. From 2011 to 2020, the global Moran's I passed the significance test at the level of 1%, and the Z values were all greater than 2.58, indicating that there was a significant spatial positive correlation and spatial aggregation effect in China's provincial grain supply chain resilience. During the study period, the overall Moran index declined slightly, indicating that the provincial correlation degree of China's grain supply chain is weakening, which is also the focus of polarization in the region and in China's space. The region must first strengthen the resilience of its own grain supply chain to ensure the stability of the grain industry under various disturbances.

**Table 4.** The global Moran's I of China's grain supply chain resilience from 2011 to 2020.

| Year | Moran's I | Z Value | p Value |
|------|-----------|---------|---------|
| 2011 | 0.334 | 4.277 | 0.000 |
| 2012 | 0.340 | 4.366 | 0.000 |
| 2013 | 0.271 | 3.556 | 0.000 |
| 2014 | 0.262 | 3.466 | 0.000 |
| 2015 | 0.218 | 2.971 | 0.001 |
| 2016 | 0.219 | 2.982 | 0.001 |
| 2017 | 0.209 | 2.856 | 0.002 |
| 2018 | 0.253 | 3.377 | 0.000 |
| 2019 | 0.249 | 3.310 | 0.000 |
| 2020 | 0.241 | 3.221 | 0.001 |

4.2.2. Local Spatial Autocorrelation

With the help of local Moran's I, the representative years are selected to interpret the local spatial similarities or differences in the development of China's grain supply chain resilience, as shown in Figure 4.

It can be seen that China's grain supply chain resilience shows the development situation of high-high agglomeration and low-low agglomeration as a whole, and forms the polarization of spatial distribution, which, once again, verifies the characteristics of decreasing spatial distribution from east to west. According to the evolution of the period, the provinces with relatively weak resilience in low-high agglomeration areas urgently need to vigorously promote high-quality development of the grain industry. In addition, the promotion effect is bound to be obvious due to the neighboring communication in advantageous areas. The western region is composed of weak resilience areas concentrated in Gansu, Qinghai and Tibet. In addition, the resilience index of Sichuan Province cannot drive the improvement of neighboring areas, which slows down the growth rate. Laying a solid foundation for the basic functions of grain supply chain is the first priority to effectively optimize resilience.

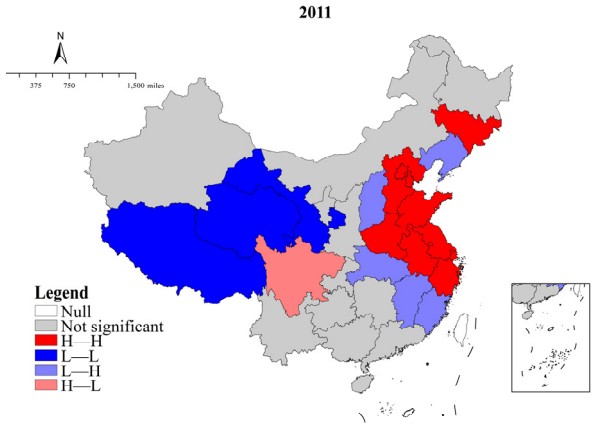

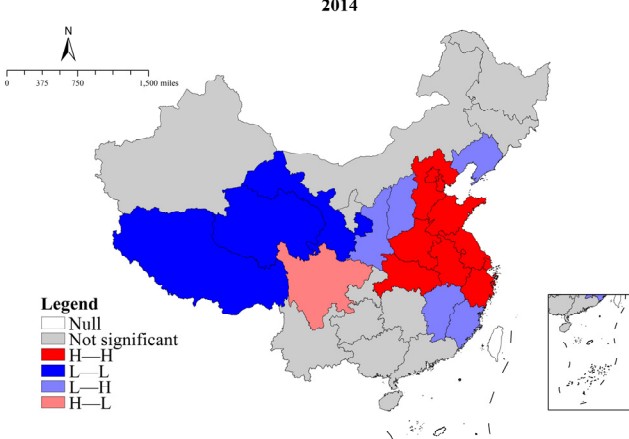

(**a**) High/low clustering characteristics in 2011

(**b**) High/low clustering characteristics in 2014

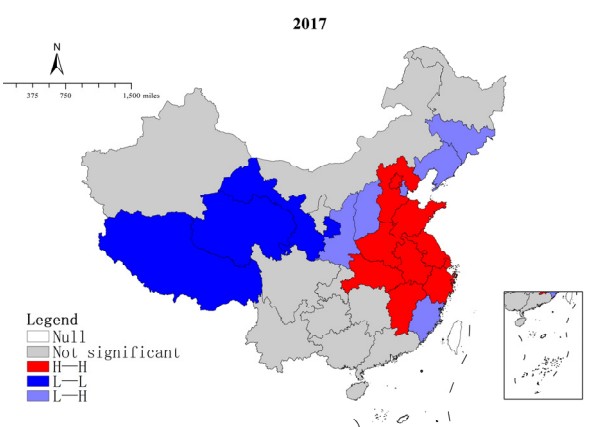

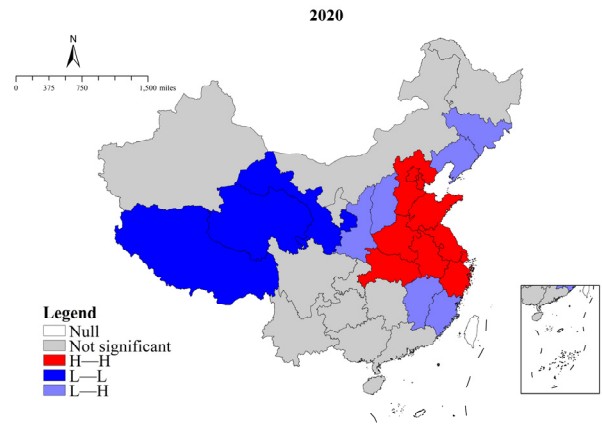

(**c**) High/low clustering characteristics in 2017

(**d**) High/low clustering characteristics in 2020

**Figure 4.** High/low clustering of China's grain supply chain resilience in 2011–2020.

## 5. Influence Factors of China's Grain Supply Chain Resilience

### 5.1. Selection of Influence Factors

The effective driving force of grain supply chain resilience must be based on risk avoidance of the grain supply chain and high-quality development of the grain industry. Ding Dong and others summed up that the basic characteristics of China's grain supply chain risk management are safety, quality, finance, reform, talents and trade [27]. Gao Weilong and others judged that factor quality, factor structure, natural environment, institutional environment and economic environment were the innovation-driven influence paths for high-quality development of the grain industry [28]. Based on these studies, the exogenous drivers are as follow: Factor quality is based on Huang Jianbai, and was measured by educational innovative human capital and investment innovative human capital [29]. Structural upgrading relied on Luo Guangqiang and other research conclusions: digital inclusive finance promoted the transformation, upgraded the agricultural industrial structure in major grain producing areas by promoting the rationalization [30], and the evaluation results of the research group of Peking University Digital Finance Research Center were applied. The institutional environment is measured by government supports, which is the sum of financial budget expenditures, i.e., agriculture, forestry and water affairs expenditures; grain and oil storage expenditures; transportation expenditures; and science and technology expenditures. The economic environment is measured by the level of economic growth and urbanization rate, that is, the per capita GDP of the current year

and the proportion of urban population at the end of the year to the total population of the region at the end of the year.

### 5.2. Effect of Grain Supply Chain Resilience

According to the test results of variance expansion factor (VIF) (abbreviated in the table), the VIF values are all less than 10, and therefore, it is concluded that there is no multicollinearity among the selected driving factor index data. Based on this, we construct a spatial Dubin model. In order to prevent possible errors in testing spatial spillover by point estimation, in this paper, we decompose the spillover effect and obtain direct effect, indirect effect and total effect. The LM test shows that the *p*-values of the spatial error model and the spatial lag model are significant at the 1% level; therefore, it is more reasonable to choose the SDM model combining the SEM model and the SAR model. The Hausman test passed the 1% significant level, and the fixed effect model was selected. The Wald test and the LR test both reject the original hypothesis that the SDM model can degenerate into SAR and reject the original hypothesis that the SDM model can degenerate into SAR at the level of 1%; therefore, the SDM model is selected. From the above analysis, it is concluded that the spatial difference of resilience index of China's grain supply chain is solidified, and the slow development of the western region leads to a deepening of the contradiction of spatial differentiation. Therefore, adopting the fixed individual model will weaken the influence of influencing factors on individuals, resulting in the problem that influencing factors cannot pass the significance test. At the same time, according to the test of the optimal effect model, the SDM model with time fixed effect is the best choice, which proves that the individual fixed effect will weaken the overall regression result. Because the regression coefficient of the spatial Dubin model cannot accurately analyze the influence of explained variables and spatial spillover, its decomposition model is the focus of this paper. The results of the spatial Dubin model are summarized in Table 5.

**Table 5.** Estimation results of the spatial Dubin model and its decomposition model.

| Variable | Main Effect Term | Spatial Conduction Effect | Direct Effect | Indirect Effect | Total Effect |
|---|---|---|---|---|---|
| lnhr [1] | 0.036 *** (0.006) | −0.001 (0.013) | 0.039 *** (0.006) | 0.028 *** (0.017) | 0.066 *** (0.019) |
| dif [1] | −0.066 *** (0.019) | 0.000 (0.014) | −0.068 *** (0.019) | −0.057 * (0.033) | −0.125 ** (0.043) |
| lngov [1] | 0.043 *** (0.010) | −0.017 (0.023) | 0.043 *** (0.011) | 0.009 (0.045) | 0.052 (0.051) |
| lnec [1] | 0.039 *** (0.015) | −0.001 (0.011) | 0.040 *** (0.015) | 0.032 (0.028) | 0.072 * (0.038) |
| urban [1] | −0.188 *** (0.047) | 0.182 ** (0.080) | −0.178 *** (0.048) | 0.155 (0.140) | −0.023 (0.162) |
| $\rho$ | | | 0.484 *** (0.097) | | |
| sigma2_e | | | 0.001 *** (0.000) | | |
| Hausman test | | | 16.25 *** | | |
| SDM-SAR | | | 19.09 *** | 18.35 *** | |
| SDM-SEM | | | 20.91 *** | 16.96 *** | |
| Both-Ind | | | 12.250 | | |
| Both-Time | | | 530.630 *** | | |
| $R^2$ | | | 0.174 | | |
| Log-likelihood | | | 564.662 | | |
| Observation | | | 310 | | |

[1] The values in brackets are the standard errors of each coefficient; ***, **, * show the significance of 1%, 5% and 10%, respectively. lnhr is the logarithm of innovative human capital; dif is the digital inclusive finance; lngov is the logarithms government supports; lnec is the logarithm of economic development level; urban is the urbanization rate.

According to the results of the spatial Dubin model, the fitting degree and reliability of the model are high, and the spatial auto-regressive coefficient $\rho$ is significant at the 1% level and its coefficient is 0.484, which shows that the resilience of the grain supply chain itself has positive spatial spillover effect. Government supports, economic development level and innovative human capital have positive driving effects, while digital inclusive finance and urbanization rate have negative driving effects. We find that strengthening government

assistance, promoting regional economic development and injecting innovative talents could quickly strengthen the grain supply chain resilience and efficiently promote some provinces with weak grain supply chain resilience. The reverse impact of digital inclusive finance shows that the depths of digital villages, digital industries and digital inclusion are insufficient. The negative correlation between urbanization rate and resilience of grain supply chain proves that China is currently in a critical period of urban-rural integration development, and it is necessary to stick to the red line of 1.8 billion mu of cultivated land to ensure grain production capacity and output. In addition, only urbanization shows spatial transmission effect, and neighboring provinces have a positive guiding effect on the resilience of local grain supply chains, which proves that provinces with advantages in grain production capacity have a radiation effect for provinces with rapid urbanization process.

Based on the decomposition of spatial spillover effect, it can be concluded that all driving factors have direct effects on the grain supply chain. Innovative human capital and digital inclusive finance also show indirect effects, which shows that the corresponding driving factors have a radiation effect, and enhancing talent allocation and digital inclusive construction can promote efficient coordination and common promotion among regions. From the total effect, it can be seen that government support and regional urbanization have a strong force on the interior of the province, and the regulation effect of "visible hand" is the first choice to improve the weak and resilient provinces. Meanwhile, it also shows that the model of strong government guidance and urbanization provinces helping the food supply chain is worth learning from.

### 5.3. Regional Heterogeneity Analysis

According to the results of the regional heterogeneity analysis (Table 6), innovative human capital, digital inclusive finance and urbanization have significant impacts on the grain supply chain in northeast China, which indicates that the transformation and development of regional grain industry is urgent, while government supports and regional economic growth still need to be strengthened to strengthen the resilience of the grain supply chain. The results in the eastern region show that the resilience of the grain supply chain is positively correlated with the level of urbanization and economic growth, and also shows the reasons for the leading position in the eastern region. In the middle region, innovative human capital and digital inclusive finance will be injected into the development of the grain industry, which will inject vitality into the learning ability and transformation ability of the grain industry, thus, promoting spatial coordination between the middle region and other regions. However, the spatial regression results in the western region explain the urgent demand for talent, and other influencing factors are not significant, which means that the infrastructure construction of the grain supply chain in the western region has significant room for improvement. It is necessary to consolidate and improve the internal control areas from the macro level, in order to accelerate the growth rate of the resilience of the grain supply chain and absorb innovative talents to seek the improvement strategy and development path of the main functional links of the grain industry.

**Table 6.** Regional heterogeneity analysis.

| Variable | Northeast | Eastern | Middle | Western |
|---|---|---|---|---|
| lnhr | 0.119 ** (0.046) | −0.049 (0.041) | 0.175 *** (0.044) | 0.031 ** (0.013) |
| dif | −0.111 *** (0.067) | −0.037 (0.041) | 0.101 * (0.054) | −0.013 (0.023) |
| lngov | 0.001 (0.028) | −0.011 (0.025) | 0.012 (0.035) | −0.006 (0.011) |
| lnec | 0.059 (0.042) | 0.073 * (0.042) | 0.079 (0.055) | −0.011 (0.018) |
| urban | 1.444 *** (0.691) | 0.456 *** (0.134) | −0.768 (0.727) | 0.192 (0.176) |
| Provincial fixation effect | YES | YES | YES | YES |
| $R^2$ | 0.759 | 0.484 | 0.291 | 0.474 |
| Observation | 30 | 100 | 60 | 120 |

The values in brackets are the standard errors of each coefficient; ***, **, * show the significance of 1%, 5% and 10%, respectively.

## 6. Discussion

In contrast to other studies, we construct an optimization study of China's grain supply chain resilience system at both the internal level and the external level. For the internal system, six dimensions of prevention, prediction, digestion, recovery, learning and transformation are selected to evaluate grain production, storage, processing, transportation and consumption resilience of the grain supply chain in each province of China, and we empirically derive the grain supply chain resilience index. The CRITIC-EWM combined evaluation is applied to improve the rigor of the assessment, to reveal the differences in indicator weights and to analyze the internal development bottlenecks and obstacles; kernel density estimation and Dagum Gini coefficient are selected to demonstrate spatio-temporal evolutionary characteristics and spatial differences of sustainable quality development. The spatial Dubin model is used to analyze the influence of exogenous driving factors to provide theoretical support for the selection of strategies to effectively promote grain supply chain resilience and to break through the bottleneck of high-high and low-low agglomerations. The findings strongly suggest that focusing on regional heterogeneity to strengthen the digestion, recovery and learning capacity of the grain supply chain is key to improving the internal system resilience. To enhance the level of grain reserve, new varieties quality, emergency security, mechanization and informatization could help to resist the sustainability of the main functions under uncertain shocks. Therefore, we propose several countermeasures.

### 6.1. Government Supports Promote the Synergy of Regional Grain Supply Chain Resilience

Government supports significantly contribute to the improvement of grain supply chain resilience. They can effectively solve the regional obvious problems and can break through the bottleneck of solidifying regional differences for China's grain supply chain resilience. Through strategy implementation, policy formulation, financial expenditures and government guidance targeted optimization can be achieved based on China's regional characteristics, to balance the layout of food security, maximize the radiation and driving effects of superior production capacity areas, and increase government financial investment to improve the infrastructure of the grain supply chain in each region. In addition, there is an urgent need for the government to enhance the food market supervision and emergency handling capability. A combination of the "visible hand" and the "invisible hand" must be used in order to strengthen the foundation of grain supply chain resilience.

### 6.2. Introduce the Innovative Human Capital for Grain Supply Chain Optimization

This study exposes the drawbacks in China's grain supply chain which is still mainly traditional and transitional, and high cost and low efficiency are the outstanding problems that hinder high-quality development of the grain supply chain. Introducing innovative human capital could help to improve the learning capacity of the grain supply chain, especially to achieve a benign development situation of timely application of new varieties, new equipment, new technologies and efficient integration of new periods, new developments and new modes. Accelerating the transformation and development of the grain supply chain to modernization would contribute by improving the suitability of the supply side and demand side of grain.

### 6.3. Digital Economy Helps the Sustainability of Grain Supply Chain Functions

China's grain supply chain is characterized by the integration of primary, secondary and tertiary industries. The digital economy with its substitution, penetration and synergy effects, can enhance the stability and sustainability of grain production, purchase, storage, processing and consumption, accelerate industrial integration to strengthen the robustness of the system and to guarantee the continued operation of core functions under shocks. Thus, the digital economy is the best driver. The integration of information, technology and innovation with the main functions of the grain supply chain can facilitate the transfor-

mation of the grain supply chain, value chain, and industry chain, and can promote higher development of food security.

## 7. Conclusions

(1) From the perspective of temporal and spatial dynamic evolution, the resilience index of China's grain supply chain is generally low and the growth rate is slow. The root cause is the traditional cropping mode that overdraws natural resources, seriously hindering the grain supply chain resilience. China's grain supply chain was apparently resilient under the COVID-19 pandemic; resilient provinces accounted for 70.97% of the grain production, reflecting the effectiveness of the food crop production strategy based on farmland management and technological application. It is found that the areas with superior grain production capacity have strong adjacent radiation force to regional grain supply chain resilience.

(2) From the perspective of spatial unit differences and spatial autocorrelation in the four regions, it shows that the spatial differences of China's grain supply chain resilience are solidified, which is manifested in the fact that inter-regional differentiation is the main spatial contradiction point. There is a positive spatial correlation between provinces in China, but the correlation degree is decreasing year over year, which also fully explains why the grain supply chain resilience gap between provinces in China is solidified. The weakening of the regional connection inevitably leads to a hierarchy of development, and the grain supply chain resilience gradually forms a trend of "high-high" and "low-low" clusters.

(3) From the influencing factors and spatial spillover effects, it is concluded that resilience optimization of the grain supply chain in China, at present, must be guided by government supports, economic development level and innovative human capital. At present, for the bottleneck of "digital inclusion" development of small farming households and micro enterprises, sticking to the red line of 1.8 billion mu of cultivated land and accelerating informatization, mechanization and large-scale production and processing are prerequisites to ensure sustainable supply. Policy implementation should introduce valuable external drivers according to the characteristics of each region's development.

Due to limitations in data availability, there are relatively few exogenous drivers of grain supply chain resilience, and adding drivers can help to increase the specificity of policy formulation. In addition, this research focused on the domestic stability of China's food security strategy, and expanding the international perspective of this study has the potential to increase the usefulness of the study. Future research will extend from grain supply chain resilience to food supply chain resilience, based on the "Great Food Outlook" to ensure food security sustainability.

**Author Contributions:** Writing—original draft and methodology, J.C.; writing—review and funding acquisition, H.J. All authors have read and agreed to the published version of the manuscript.

**Funding:** This research was funded by the Jilin Provincial Social Science Foundation Project "The realization path and policy design of digital economy to promote farmers' income in Jilin Province" (2022B050179).

**Institutional Review Board Statement:** Not applicable.

**Informed Consent Statement:** Not applicable.

**Data Availability Statement:** Not applicable.

**Conflicts of Interest:** The authors declare no conflict of interest.

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
