# Peer review of "Spatio-Temporal Differentiations and Influence Factors in China’s Grain Supply Chain Resilience"

_sustainability, doi:10.3390/su15108074_

Round 1
Reviewer 1 Report
Authors have developed rather interesting research on an urgent topic. The paper is relatively well structured and based on comprehensive analytical data. Nevertheless, there are some recommendations that might improve the quality of the paper:
1) In lines 51-67 of section 1, different approaches to understanding resilience are described. Expanding this part of Section 1 by clarifying the components of system resilience and authors’ definitions of these components is advisable. It might help to highlight more common features of resilient systems;
2) Expanding part of Section 1 concerning food industry and supply chain resilience is also recommended. Otherwise, the current shape of this part of Section 1 (lines 68-72) is not sufficient;
3) in general Introduction section does not support the research objectives. It is needed to identify research gaps clearly, point out those discussion questions that will be researched in the paper and highlight how the realization of the research objectives might help to cover the existing research gap and to which extent;
4) it is recommended to add a literature review section and construct it with add-ins mentioned in 1-3 of the review report and partly with some pieces of research design block;
5) implementation of the literature review section also might clarify the choice of the indicators of the grain supply chain resilience evaluation index system from Table; it also needs more clarification on the procedure of the indicators’ selection and methods of weighting coefficients identification; moreover, information from Table 1 better illustrates results but not the methodology of the Index construction, it should be relocated to the sufficient block of the paper;
6) it is also recommended to reconstruct the Methodology section with its closer correlation with the research objectives: to point out primarily the theoretical objective and secondary to describe the method that will be applied to reach this objective but not wise versa;
7) it is better to use, for example, “sufficient resilience” (or other relevant characteristics) instead of “slightly strong resilience” (line 214);
8) the Conclusion section might demonstrate new research findings compared to the existing research.
In general, the research is rather interesting and valuable but should be improved for better transparency and understanding for inexperienced readers.
Minor editing of English language required
Author Response
Dear Reviewer:
I would like to thank the respected Reviewers for their constructive comments on our manuscript “sustainability-2355869”. I have considered the comments very carefully and have revised the paper accordingly. All changes to the text and figures are shown. Thanks to the Reviewers. Firstly, in order to enhance the quality of research, the abstract, introduction and conclusion sections, which were commonly pointed out by the reviews, have been overhauled in this paper; Secondly, considering the research logic, we added a sixth section—Discussion, which to sort out the research process and give effective strategies based on the empirical results. Some parts of the paper have been refined towards the research design and framework; Finally, professional translations have been made for the English presentation in the paper. The revised contents have been improved and marked in the paper according to the Reviews' opinions.
Comment 1: In lines 51-67 of section 1, different approaches to understanding resilience are described. Expanding this part of Section 1 by clarifying the components of system resilience and authors’ definitions of these components is advisable. It might help to highlight more common features of resilient systems;
Author’s answer: Thanks for the specific suggestions given for the writing of the introduction section, the background and literature review sections are written according to Comment 1, which in order to fully improve the theoretical basis of the study.
Comment 2: Expanding part of Section 1 concerning food industry and supply chain resilience is also recommended. Otherwise, the current shape of this part of Section 1 (lines 68-72) is not sufficient;
Author’s answer: The addition of supply chain resilience, agricultural resilience concept elaboration and evaluation is helpful to draw out the research design ideas of this paper and help articulate more clearly. (Line 66-72; Line 86-91.)
Comment 3: in general Introduction section does not support the research objectives. It is needed to identify research gaps clearly, point out those discussion questions that will be researched in the paper and highlight how the realization of the research objectives might help to cover the existing research gap and to which extent;
Author’s answer: Considering the Introduction section does not support the research objectives, we added the Background and Literature Review, in order to enrich the slightly weak description. We emphasize the value of this study at this stage of development, adding theoretical elaboration to show that the basis of our study is adequate.
Comment 4: it is recommended to add a literature review section and construct it with add-ins mentioned in 1-3 of the review report and partly with some pieces of research design block;
Author’s answer: According to your suggestions, we have made serious revisions to emphasize the practical significance and theoretical innovation of the research perspective of this paper, and made detailed explanations respectively, which effectively enhance the value of the research of this paper, thank you.
Comment 5: implementation of the literature review section also might clarify the choice of the indicators of the grain supply chain resilience evaluation index system from Table; it also needs more clarification on the procedure of the indicators’ selection and methods of weighting coefficients identification; moreover, information from Table 1 better illustrates results but not the methodology of the Index construction, it should be relocated to the sufficient block of the paper;
Author’s answer: The literature review was written with the idea of extending the definition of resilience to the definition of grain supply chain resilience and the evaluation of resilience to the evaluation of food supply chain resilience, respectively, to illustrate the theoretical research basis of the article. Meanwhile, there are relatively few existing empirical articles on the evaluation of grain supply chain resilience, reflecting the academic contribution of this paper's research. We added the part where the weight of the indicator was increased.(3.1. Line246-251) An elaboration of the weighting results was added to the paper to illustrate how reinforcement occurs within the system of grain supply chain resilience. (3.1. Line 252-262)
Comment 6: it is also recommended to reconstruct the Methodology section with its closer correlation with the research objectives: to point out primarily the theoretical objective and secondary to describe the method that will be applied to reach this objective but not wise versa;
Author’s answer: The section on methodology is to show the research methods applied in this paper and how to conduct the evaluation more scientifically? How to reflect the spatio-temporal evolution? How to reflect spatial differences? How to choose effective promotion measures? The literature review section illustrates the idea of the study by way of theoretical and empirical values. Therefore, due to the word count of the article, this section is a direct listing of the methods. (1.2 line 106-112)
Comment 7: it is better to use, for example, “sufficient resilience” (or other relevant characteristics) instead of “slightly strong resilience” (line 214);
Author’s answer: I really appreciate your excellent idea, I thought of so many words to describe it while writing the paper, but none of them felt appropriate.
Comment 8: the Conclusion section might demonstrate new research findings compared to the existing research.
Author’s answer: The conclusion section of the paper was carefully revised to summarize and demonstrate new research findings compared to the existing research from three aspects to improve the research value of the paper, and to add research gaps and research outlook to increase research paradigms. (7.Conclusions line 534-568)
Meanwhile, we thoroughly edited the English translation of the article to improve the grammar and terminology. Thank you very much for your approval of this paper. This paper has gone through many revisions from understanding the concept of food supply chain resilience, evaluation index system construction, data collection, method selection to writing the paper to enhance the innovation of the research. Your valuable suggestions have added academic and value to this paper. Through the revisions, the research of the paper is more enriched and reasonable, and it has improved the readability of the paper, which is what I need to improve when I focus on writing the paper, thank you!
Best wishes!
Jinrui Chang

Reviewer 2 Report
Rewrite the Abstract.
Focus on contribution of this study
Rewrite the Introduction section to improve quality of communication.
Strengthening of literature review section is suggested. Refer relevant and recent papers from reputed journals including Sustainability.
Cite the references in main body in professional manner as per Sustainability format.
Research design is satisfactory.
What is the contribution of this study?
Highlight the novelty of this study.
Rewrite the Conclusion accordingly
Proofreading of entire manuscript is suggested. The statistical calculations need to be rechecked.
Author Response
Dear Reviewer:
I would like to thank the respected Reviewers for their constructive comments on our manuscript “sustainability-2355869”. I have considered the comments very carefully and have revised the paper accordingly. All changes to the text and figures are shown. Thanks to the Reviewers. Firstly, in order to enhance the quality of research, the abstract, introduction and conclusion sections, which were commonly pointed out by the reviews, have been overhauled in this paper; Secondly, considering the research logic, we added a sixth section—Discussion, which to sort out the research process and give effective strategies based on the empirical results. Some parts of the paper have been refined towards the research design and framework; Finally, professional translations have been made for the English presentation in the paper. The revised contents have been improved and marked in the paper according to the Reviews' opinions.
Comment 1: Rewrite the Abstract.
Author’s answer: The abstract section has been iterated and carefully revised to present the valuable findings of this paper. The abstract is written along the lines of the paper's research methodology, significance and findings.(Line 8-18)
Comment 2: Focus on contribution of this study.
Author’s answer: We thank you for your suggestions and we have carefully revised it. We have presented the research innovations and implications of the article through different perspectives in the background and literature review sections.(Line 48-55 and Line 99-106)
Comment 3:Rewrite the Introduction section to improve quality of communication.
Author’s answer: Considering the Introduction section does not support the research objectives, we added the Background and Literature Review, in order to enrich the slightly weak description.
Comment 4: Strengthening of literature review section is suggested. Refer relevant and recent papers from reputed journals including Sustainability.
Author’s answer: The literature review was written with the idea of extending the definition of resilience to the definition of grain supply chain resilience and the evaluation of resilience to the evaluation of food supply chain resilience, respectively, to illustrate the theoretical research basis of the article. Meanwhile, there are relatively few existing empirical articles on the evaluation of grain supply chain resilience, reflecting the academic contribution of this paper's research. We also cited relevant research papers from Sustainability. (1.2. Line 57-108)
Comment 5: Cite the references in main body in professional manner as per Sustainability format.
Author’s answer: Following the sustainable format, we modified the article to cite references in the text in a professional manner. Meanwhile, we have carefully revised the content of the article as a way to improve the standardization of the article.
Comment 6: What is the contribution of this study?
Author’s answer: The research contributions of this paper are presented in the last paragraph of the background section and in the last paragraph of the literature review. The theoretical value is the deconstruction of the grain supply chain system by main functions and the visualization of resilience capacity dimensions, which improves the accuracy of the system resilience evaluation, clarifies the weaknesses of the internal system resilience and enhances the targeting of the research. The empirical value is that the study is based on Chinese provinces, and the analysis of spatial and temporal characteristics can provide suggestions for synergistic sustainable development, introduce external drivers to measure effective impact scenarios, and provide references for strengthening the implementation of regional grain supply chain resilience policies. This study forms a two-dimensional internal and external scenario for improving the grain supply chain resilience.
Comment 7: Highlight the novelty of this study.
Author’s answer: This paper once again emphasized the specificity of our study, compared to other scholars' evaluations of systemic resilience.(2.1 Line 125-127) (7. Discussion Line 469-487)
Comment 8: Rewrite the Conclusion accordingly
Author’s answer: Proofreading of entire manuscript is suggested. The statistical calculations need to be rechecked. Thanks for your specific advice. The conclusion section of the paper was carefully revised to summarize and demonstrate new research findings compared to the existing research from three aspects to improve the research value of the paper. We mentioned the limitations and future research directions in the end. The modified contents are in line 530-563 of section 7.
Comment 9:Proofreading of entire manuscript is suggested. The statistical calculations need to be rechecked.
Author’s answer: Based on your suggestions, we carefully revised the writing of each part of the article and refined the contents. Adjustments were completed in all parts, from the research framework and design to the empirical analysis. Your suggestions have greatly improved the quality of the article and I noticed my shortcomings in writing clearly, and I will strengthen my study to improve my writing skills.
Meanwhile, we thoroughly edited the English translation of the article to improve the grammar and terminology. Thank you very much for your approval of this paper. This paper has gone through many revisions from understanding the concept of food supply chain resilience, evaluation index system construction, data collection, method selection to writing the paper to enhance the innovation of the research. Your valuable suggestions have added academic and value to this paper. Through the revisions, the research of the paper is more enriched and reasonable, and it has improved the readability of the paper, which is what I need to improve when I focus on writing the paper, thank you!
Best wishes!
Jinrui Chang

Reviewer 3 Report
The author has demonstrated a significant level of scientific rigour throughout the manuscript. Nonetheless, I have a few minor comments about how this manuscript could be enhanced:
Point 1: It is strongly suggested that authors employ sources of figures throughout the entirety of the Manuscript.
Point 2: It is advised that authors complete their bibliographies by citing the following databases: Lines no 122 to 128.
China Statistical Yearbook, China Grain Yearbook, China Grain and Material Reserve Yearbook, China Rural Statistical Yearbook, China Science and Technology Statistical Yearbook, China Agricultural Products Processing Yearbook, China Population and Employment Statistical Yearbook, Provincial Statistical Yearbook, Provincial Rural Statistical Yearbook, China Internet Development Statistical Report, Brick Agricultural Database, the website of the Ministry of Agriculture and Rural Affairs of the People's Republic of China
Point 3: It is standard practice to mention any limitations imposed by the research in the conclusion. Additionally, future research directions should be included.
Minor editing of English language required
Author Response
Dear Reviewer:
I would like to thank the respected Reviewers for their constructive comments on our manuscript “sustainability-2355869”. I have considered the comments very carefully and have revised the paper accordingly. All changes to the text and figures are shown. Thanks to the Reviewers. Firstly, in order to enhance the quality of research, the abstract, introduction and conclusion sections, which were commonly pointed out by the reviews, have been overhauled in this paper; Secondly, considering the research logic, we added a sixth section—Discussion, which to sort out the research process and give effective strategies based on the empirical results. Some parts of the paper have been refined towards the research design and framework; Finally, professional translations have been made for the English presentation in the paper. The revised contents have been improved and marked in the paper according to the Reviews' opinions.
Point 1: It is strongly suggested that authors employ sources of figures throughout the entirety of the Manuscript.
Author’s answer: Thank you very much for your suggestion, we considered the word count of the article did not specify before. For the data sources we have made detailed changes to explain each index and accounting methods. The explanation of the indexes in Table 1 also explains how the indexes were calculated. The modified contents are in lines 134-154 of section 2.
Point 2: It is advised that authors complete their bibliographies by citing the following databases:
Author’s answer:I really appreciate your examples of these databases that help improve the accuracy of writing the data sources section of our papers. This section has been carefully revised to improve the drawability of our research. The modified contents are in lines 134-154 of section 2.
Point 3: It is standard practice to mention any limitations imposed by the research in the conclusion. Additionally, future research directions should be included.
Author’s answer: Thanks for your specific advice. The conclusion section of the paper was carefully revised to summarize and demonstrate new research findings compared to the existing research from three aspects to improve the research value of the paper. We mentioned the limitations and future research directions in the end. The modified contents are in lines 544-550 of section 7.
Meanwhile, we thoroughly edited the English translation of the article to improve the grammar and terminology. Thank you very much for your approval of this paper. This paper has gone through many revisions from understanding the concept of food supply chain resilience, evaluation index system construction, data collection, method selection to writing the paper to enhance the innovation of the research. Your valuable suggestions have added academic and value to this paper. Through the revisions, the research of the paper is more enriched and reasonable, and it has improved the readability of the paper, which is what I need to improve when I focus on writing the paper, thank you!
Best wishes!
Jinrui Chang

Reviewer 4 Report
Can you explain anti-globalization trend? Where do you see this trend or which scholar has proved?
Please, make background to your research design and figure. All the elements of the figure should be justified by previous literature. Please take into consideration this article: DOI: 10.2478/euco-2020-0014. In the description of the figure, add the citations. Which scholars are proving the facts that you have included in your figure.
The source of the table 1 is not clear. Where this data comes from? Or how they are calculated? Please explain in the text before or directly after of the table.
Please compare the results with the previous studies’ outcomes.
Please explain well the implication of this study. Also, describe limitations and future research directions.
Author Response
Dear Reviewer:
I would like to thank the respected Reviewers for their constructive comments on our manuscript “sustainability-2355869”. I have considered the comments very carefully and have revised the paper accordingly. All changes to the text and figures are shown. Thanks to the Reviewers. Firstly, in order to enhance the quality of research, the abstract, introduction and conclusion sections, which were commonly pointed out by the reviews, have been overhauled in this paper; Secondly, considering the research logic, we added a sixth section—Discussion, which to sort out the research process and give effective strategies based on the empirical results. Some parts of the paper have been refined towards the research design and framework; Finally, professional translations have been made for the English presentation in the paper. The revised contents have been improved and marked in the paper according to the Reviews' opinions.
Comment 1:Can you explain anti-globalization trend? Where do you see this trend or which scholar has proved?
Author’s answer: The elaboration of counter-globalization in this paper is simply to highlight the background of the current uncertainty shocks that are disturbing world food security. The expression "anti-globalization" draws on the statement by Shenggen Fan(2021) "the international geopolitical and economic environment has undergone profound changes in recent years, and the intensification of counter-globalization and trade protectionist measures has led to changes in the global food trade pattern and increased the risk of China's rational use of food resources in the international market. " To avoid ambiguity, this is deleted in this paper.
Comment 2: Please, make background to your research design and figure. All the elements of the figure should be justified by previous literature. Please take into consideration this article: DOI: 10.2478/euco-2020-0014. In the description of the figure, add the citations. Which scholars are proving the facts that you have included in your figure.
Author’s answer: The inspiration for this paper is based on a report published by the FAO in 2021 this paper decomposed resilience into prevention, anticipation, digestion, recovery, learning and transformation capabilities drawing on the FAO definition of resilience(the report in 2021). FAO pointed out that a truly resilient agricultural food system must have solid ability of prevention, prediction, digestion, adaptation and transformation, which was cited in the literature review(2.2.). The exogenous drivers was based on some research, which was cited in the selection of influence factors(5.1.). We focused on the characteristics of China's grain supply chain and expanded evaluation capacity and exogenous factors of system resilience. Based on the analysis, we drew the research framework. Therefore, the design of the research framework is based on top of existing theories and we designed it based on our own research. Figure 1 is used to demonstrate the research logic of this paper. Therefore, I would also like to refer to your suggestion that this framework source has a borrowed part as well as an innovative part, is it added here in the source?
Comment 3:The source of the table 1 is not clear. Where this data comes from? Or how they are calculated? Please explain in the text before or directly after of the table.
Author’s answer: Thank you very much for your suggestion, we considered the word count of the article did not specify before. For the data sources we have made detailed changes to explain each index and accounting methods. The explanation of the indexes in Table 1 also explains how the indexes were calculated. The modified contents are in line 134-154 of section 2.2.
Comment 4: Please compare the results with the previous studies’ outcomes.
Author’s answer: During the writing of the article, I also reflected on the fact that there are shortcomings in the writing of the abstract, introduction, and conclusion of this paper. Thanks to your valuable comments, the introduction section introduces the background of the study and the theoretical basis of the existing research, which lays the foundation for highlighting the significance and value of the research in this paper. A limitation of existing scholars for the theoretical research part to go in conjunction with macro studies lies in data acquisition, we spent a lot of time in selecting indicators and accounting. The foreign scholars focusing more on the optimization of micro instances and domestic scholars mainly on qualitative research on mechanisms and theories. It is worth emphasizing that the papers on grain supply chain resilience is relatively inadequate. The modified contents are in line 48-56 of section 1.1, in line 97-109 of section 1.2 and in line 464-482 of section 6.
Comment 5: Please explain well the implication of this study. Also, describe limitations and future research directions.
Author’s answer: The research in this paper empirically analyzes the current situation of provincial grain supply chain resilience in China, scientifically dissects the endogenous bottlenecks and efficient exogenous drivers of grain supply chain development, and provides an effective reference for promoting the synergistic development of regional grain supply chains and improving the grain supply chain resilience in China. The conclusion section of the paper was carefully revised to summarize and demonstrate new research findings compared to the existing research from three aspects to improve the research value of the paper, and to add research gaps and research outlook to increase coherence. (7.Conclusion Line 560-566)
Meanwhile, we thoroughly edited the English translation of the article to improve the grammar and terminology. Thank you very much for your approval of this paper. This paper has gone through many revisions from understanding the concept of food supply chain resilience, evaluation index system construction, data collection, method selection to writing the paper to enhance the innovation of the research. Your valuable suggestions have added academic and value to this paper. Through the revisions, the research of the paper is more enriched and reasonable, and it has improved the readability of the paper, which is what I need to improve when I focus on writing the paper, thank you!
Best wishes!
Jinrui Chang

Round 2
Reviewer 1 Report
Authors have developed rather interesting research on an urgent topic. All the suggestions for paper improvement are sufficiently considered by the authors.
Best wishes!
Reviewer 2 Report
I appreciate the efforts put in by the author/s in revising the manuscript.